# Cancer genome datamining and functional genetic analysis implicate mechanisms of ATM/ATR dysfunction underpinning carcinogenesis

Erik Waskiewicz[1], Michalis Vasiliou[1], Isaac Corcoles-Saez[1] & Rita S. Cha [1]✉

ATM and ATR are conserved regulators of the DNA damage response linked to cancer. Comprehensive DNA sequencing efforts identified ~4,000 cancer-associated mutations in ATM/ATR; however, their cancer implications remain largely unknown. To gain insights, we identify functionally important conserved residues in ATM, ATR and budding yeast Mec1[ATR] via cancer genome datamining and a functional genetic analysis, respectively. Surprisingly, only a small fraction of the critical residues is in the active site of the respective enzyme complexes, implying that loss of the intrinsic kinase activity is infrequent in carcinogenesis. A number of residues are solvent accessible, suggestive of their involvement in interacting with a protein-partner(s). The majority, buried inside the respective enzyme complexes, might play a structural or regulatory role. Together, these findings identify evolutionarily conserved ATM, ATR, and Mec1[ATR] residues involved in diverse aspects of the enzyme function and provide fresh insights into the elusive genotype-phenotype relationships in ATM/ATR and their cancer-associated variants.

[1] School of Medical Sciences and North West Cancer Research Institute, Bangor University, Bangor, UK. ✉email: r.cha@bangor.ac.uk

TM (*A*taxia *T*elangiectasia *M*utated) and ATR (*AT*m and *R*ad3 related) are conserved serine/threonine kinases responsible for mediating the DNA damage response (DDR)[1,2]. In humans, germline mutations in *ATM* and *ATR* lead to Ataxia-Telangiectasia (A-T) and Seckel syndrome, respectively, characterized by a constellation of symptoms, including neurodegeneration, cancer, diabetes, infertility, and microcephaly[3,4]. Somatic mutations in ATM or ATR contribute to carcinogenesis by promoting genome instability. Paradoxically, ATM and ATR are prime cancer drug targets because DDR inhibition enhances efficacy of therapeutic radiation and many chemotherapeutic agents[5,6].

ATM/ATR proteins are found in all eukaryotes, including the ATM/ATR in mammals and plants, the Tel1[ATM]/Mec1[ATR] in budding yeast, and Tel1[ATM]/Rad3[ATR] in fission yeast. These proteins belong to the phosphatidylinositol 3-kinase-related kinase (PIKK) super family, which also include mTOR and DNA-PK[7]. PIKKs are giant HEAT (*H*untington, elongation factor, *E*F3, protein phosphatase PP2*A* and *T*OR) proteins, comprising 40–50 tandem repeats of the HEAT motif followed by a highly conserved kinase domain (Fig. 1a)[8]. The HEAT repeat region, in turn, comprises three structurally conserved domains, the N-spiral/solenoid/spiral, C-spiral/bridge/pincer, and the FAT (*F*RAP-*A*TM-*T*RRAP; Fig. 1a).

Cryo-EM studies show that the human ATM/ATR and the budding yeast Tel1[ATM]/Mec1[ATR] exist as a homodimer, comprising two identical protomers connected via multiple interfaces (e.g., Fig. 1c)[9–12]. Under unchallenged conditions, the dimeric complex exists as a minimally active enzyme, in which substrate access to the active site is sterically hindered by several conserved features, including the tetratricopeptide repeat domain 3 (TRD3) in the FAT domain and the PIKK-regulatory domain (PRD) in the kinase domain. In response to stress, the complex would undergo an allosteric change, enabling access to the active site[9–12]. The activation might take place in the dimer context or entail a dimer-to-monomer transition (Fig. 1d).

The sheer size and structural complexity of ATM/ATR proteins render a comprehensive genetic or functional analysis challenging. Accordingly, several important questions remain unanswered, including the genotype–phenotype and structure–function relationships in ATM/ATR and their disease-associated variants[3,13,14]. Here, we address these questions utilizing cancer genome datamining, molecular modeling, and functional genetic analyses. Our findings provide insights into the potential functional impact of numerous ATM/ATR mutations found in cancer.

## Results

**Cancer-associated ATM/ATR missense mutations are found along the entire length of the polypeptide.** We used the cBio Cancer Genomics Portal database (cBioPortal) to collect information on ATM/ATR mutations identified in cancer (http://cbioportal.org)[15,16]. Among the 46,588 tumor samples in the "curated set of nonredundant studies", ATM and ATR are mutated in many different cancer types with the frequency ranging from ~1% for brain or testicular cancer to ~8–10% for endometrial, bladder, or colorectal cancer (Fig. 1e and Supplementary Table 1). Overall, 2551 (5.5%) and 1394 (3.0%) of the 46,588 samples carry a mutation in ATM and ATR, respectively

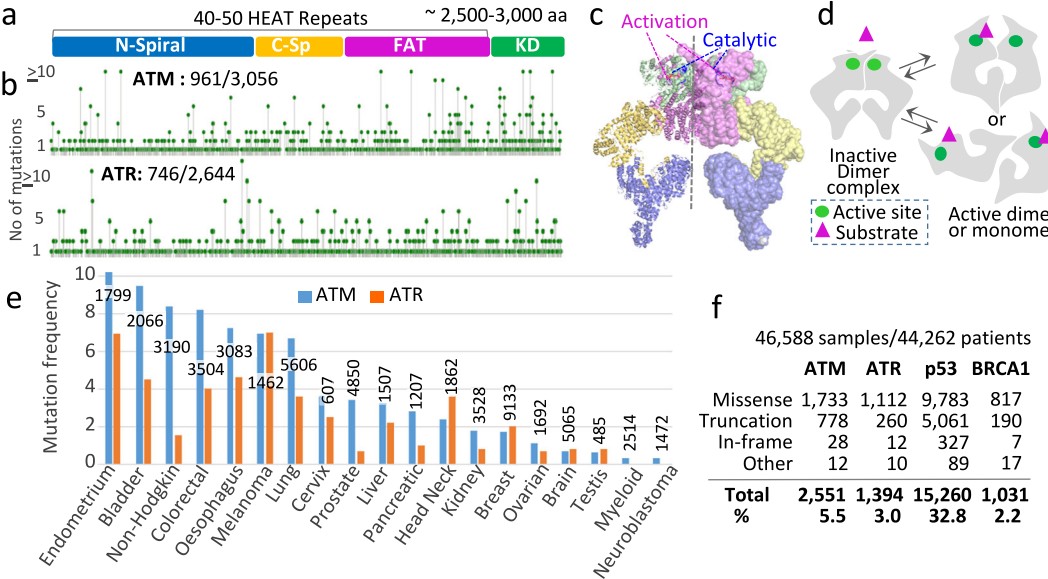

**Fig. 1 ATM/ATR mutations in human cancer. a** ATM/ATR proteins are large (~2200–3000 residues) and comprise 40–50 tandem HEAT repeats followed by a highly conserved kinase domain[45]. The HEAT repeat domain is divided into three conserved structural motifs referred to as the N-spiral/solenoid, C-spiral/bridge/pincer, and FAT domains[9–12]. **b** Screen shot images from the cBio Cancer Genomics Portal website (http://cbioportal.org). The images show the location and frequencies of ATM/ATR residues mutated in 46,588 tumor samples in the database. In ATM, 961 of the 3056 residues are mutated; in ATR, 746 of the 2644 residues are mutated (Supplementary Data 1 and 2). **c** A cryo-EM model structure of the ATM enzyme complex (PDB 5NPO, 5.70 Å)[9]. The complex is a dimer comprising two identical protomers that share multiple interfaces. The protomer on the right and left are in surface and cartoon representations, respectively. Both protomers are shown in four colors, each corresponding to the indicated domain in **a**. The catalytic and activation loops in the active site are shown in red and blue, respectively. **d** Model: the ATM/ATR and Tel1/Mec1 enzyme complexes are in dynamic equilibrium between an inactive and active conformations. Under unchallenged conditions the dimeric complex exists as a minimally active enzyme, in which substrate accessibility to the active site is sterically hindered. Stress-dependent conformational change activates the enzyme by relieving the block. The activation may take place in the dimer context or entail a dimer-to-monomer transition[9–12,46]. **e** ATM/ATR mutation frequencies in the indicated cancer types (Supplementary Table 1; http://cbioportal.org). The number of tumor samples analyzed for each cancer type is as indicated. **f** The number of cancer-associated missense, truncation, in-frame, or other mutations in *ATM*, *ATR*, *P53*, or *BRCA1* (http://cbioportal.org).

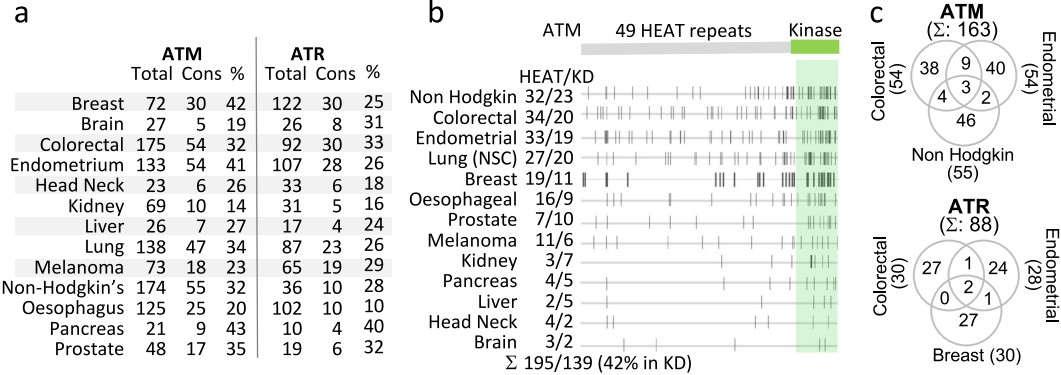

**Fig. 2 Tissue specificity of conserved ATM/ATR residues mutated in cancer. a** Extent of conservation in the ATM and ATR residues mutated in the indicated cancer type. "Total": the total number of ATM or ATR residues mutated in the indicated cancer type (Supplementary Data 1 and 2). "Cons": the number of mutated residues that are conserved in yeast Mec1 and/or Tel1 (Supplementary Data 3 and 4). "%": the fraction of conserved residues. **b** Location of the conserved ATM residues mutated in the indicated cancer type (Supplementary Data 3). The human ATM polypeptide comprises 49 HEAT units and a kinase domain. "HEAT/KD": the number of residues in the respective regions. Overall, 42% (139/336) of the conserved residues mutated in cancer are in the kinase domain. **c** Extent of overlap among the conserved ATM and ATR residues mutated in the indicated cancer type (Supplementary Data 3 and 4).

(Fig. 1f). In comparison, 15,260 (32.8%) harbor a mutation in the *P53* tumor suppressor, one of the most frequently mutated genes in cancers, and 1031 (2.2%) in *BRCA1*, another well-characterized tumor suppressor gene (Fig. 1f; http://cbioportal.org)[15,16].

In agreement with an earlier study[14], majority of the cancer-associated alterations are missense mutations leading to a codon change, followed by truncation and in-frame mutations (Fig. 1f and Supplementary Table 1). In ATM and ATR, the latter two would lead to a kinase dead phenotype due to the fact that the essential kinase domain is located at the C-terminus (e.g., Fig. 1a). On the other hand, impact of a missense mutation is difficult to ascertain because it would depend on the location and nature of the residue change.

We identified 961 ATM and 746 ATR residues that are mutated to a different residue(s) in the database (Fig. 1b, and Supplementary Data 1 and 2). A previous study, based on a smaller sample size ($n = 5402$), found that missense mutations in ATM are enriched in the kinase domain[14]. The current analysis, based on a much larger sample size ($n = 46,588$), shows that the mutated ATM and ATR residues are found along the entire length of the respective polypeptide (Fig. 1b, and Supplementary Data 1 and 2).

**ATM/ATR residues mutated in cancer are cancer type specific.** In general, cancer-associated mutations are tissue specific, reflecting the cellular processes and the types of mutagen exposure relevant for the tissue[17,18]. We wished to determine whether this was also the case for ATM/ATR missense mutations. To this end, we examined 13 different cancer types individually. We find that the number of mutated ATM/ATR residues varies widely from one cancer type to another. For example, among the 3504 colorectal cancer sample, 92 ATR residues are found to be mutated; in comparison, only 19 ATR residues were mutated among the 4850 prostate tumor samples (Fig. 1e, Fig. 2a, and Supplementary Table 1). In general, the cancers with higher ATM or ATR mutation frequencies tend to have higher number of mutated residues; for example, endometrial cancer, non-Hodgkin's lymphoma, and colorectal cancer, with an ATM mutation frequency of 8–10% (Fig. 1e), each has over 100 mutated residues (Fig. 2a and Supplementary Table 1). However, there are exceptions; for example, breast cancer has the highest number of mutated ATR residues ($n = 122$; Fig. 2a) despite a relatively modest mutation frequency (2%; Fig. 1e). It is possible that in the

latter cancer types, there are no strong mutation hotspots, resulting in an increase in the total number of mutated residues, rather than the incidence of mutations at a specific residue.

Given the inherent genome instability of cancer cells, the majority of the cancer-associated mutations are likely passenger mutations[19,20]. As a means to identify mutations that are more likely to impact the protein function, we decided to focus on those that occurred at an evolutionarily conserved residue[21]. To identify conserved residues, we utilized the Clustal-Omega multiple sequence alignment program (Supplementary Methods and Supplementary Fig. 1). At the amino acid level, the human ATM/ATR proteins share ~90%, ~65%, ~30%, and ~20% identity with orthologs of the *Mus musculus*, *Xenopus laevis*, *Arabidopsis thaliana*, and *Saccharomyces cerevisiae*, respectively. We find that 233 of the 961 ATM and 147 of the 746 ATR residues mutated in cancer are conserved in the budding yeast ATM/ATR proteins Tel1 and/or Mec1 (Supplementary Data 3 and 4). We restricted all further analyses to these 380 conserved ATM/ATR residues.

We find that the conserved ATM/ATR residues mutated in different cancer types show distinct distribution patterns along the respective polypeptides (Fig. 2b and Supplementary Fig. 2a). We performed a more detailed analysis on a few selected cancer types: for ATM, we chose colorectal cancer, endometrial cancer, and non-Hodgkin's lymphoma because each has a comparable number of mutated residues that are conserved in Mec1 and/or Tel1, 54, 54, and 55, respectively (Fig. 2a). For ATR, we chose colorectal, breast, and endometrial cancers based on the same criteria ($n = 30$, 30, and 28, respectively; Fig. 2a). Among the 163 ATM residues examined, only three are common to the three cancer types (Fig. 2c and Supplementary Data 3). Similarly, only two of the 88 ATR residues are common (Fig. 2c and Supplementary Data 4).

**Majority of the conserved ATM/ATR residues mutated in cancer localize internally.** We utilized the atomic resolution cryo-EM models of ATM (PBD 5NPO, 5.70 Å) and ATR (PBD 5YZO, 4.70 Å)[9,10] to map the mutated residues in three dimension. Figure 3a shows location of the 46 ATM residues mutated in non-Hodgkin's lymphoma, but not in colorectal or endometrial cancer (Fig. 2c and Supplementary Data 3). The model highlights a total of 92 residues, 46 on each of the two protomers (Fig. 3a): the protomer on the right is shown in a transparent cartoon representation with all 46 mutated residues visible on the

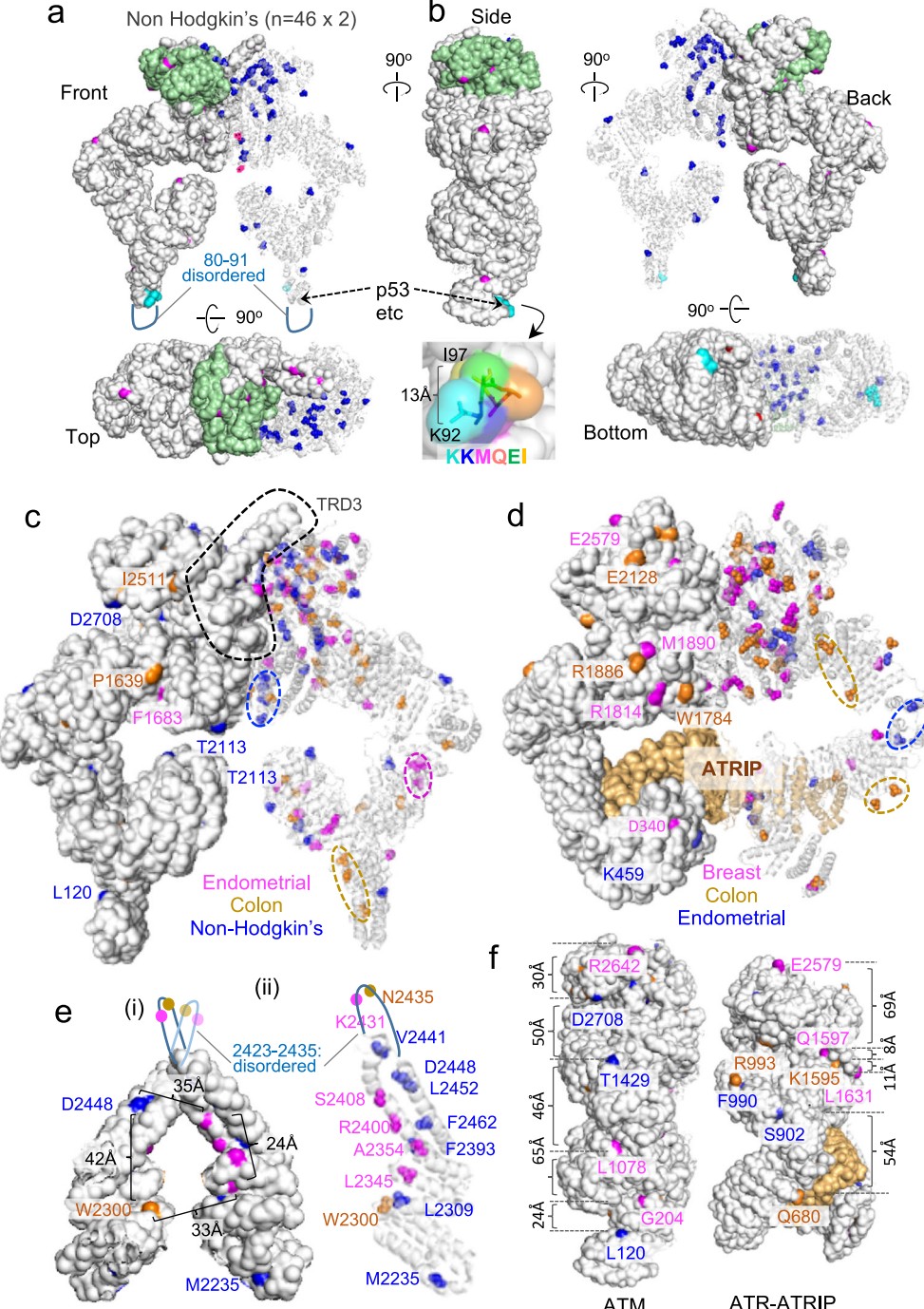

**Fig. 3 Molecular modeling of the conserved ATM and ATR residues mutated in cancer. a** The conserved 46 ATM residues mutated in non-Hodgkin's lymphoma but not in endometrial or colon cancers (Fig. 2c and Supplementary Data 3) are mapped onto a cryo-EM model of the dimeric ATM complex (PDB 5NPO, 5.70 Å)[9]. The protomer on the right-hand side is in cartoon representation and allows visualization of all 46 residues (blue). The protomer on the left-hand side is shown in surface representation to identify residues that are readily visible on the surface among the 46 (magenta). Green: kinase domain. Upper and lower panels show a view from the front and the top. Cyan: residues 92–97 involved in binding to several ATM-interacting proteins, including p53 (ref. [47]). Residues 80–91 are in a disordered region and missing in the model[9]. **b** The side, back, and bottom views of the image in **a**. Also shown are the surface representation of the region around the seven residues involved in p53 binding and the distance between the K92 and I97 in Å. **c** Same as **a** except that the complex is showing residues mutated in non-Hodgkin's lymphoma (*n* = 46; blue), endometrium cancer (*n* = 40; magenta), or colon cancer (*n* = 38; brown; Supplementary Data 3). TRD3: a regulatory motif examined in detail in **e**. Dashed oval: an area enriched for the residues mutated in the given cancer type. **d** Same as in **c**, but for ATR residues mutated in cancers of breast (*n* = 27; magenta), colon (*n* = 27; brown), or endometrium (*n* = 28; blue; Supplementary Data 4). The ATR model used: PDB 5YZO, 4.70 Å (ref. [10]). **e** A higher resolution image of the TRD3 in **c**. (i) The TRD3 motifs from both protomers are shown in surface representation to visualize solvent accessible residues among the mutated residues. The 22 residues between 2423 and 2435 are disordered and missing in the model[9]. (ii) The right-hand side TRD3 motif is shown in cartoon representation to enable visualization of all mutated residues. **f** A side view of the images shown in **c** and **d**, showing the distances between some of the solvent accessible residues.

structure. The protomer on the left is shown in a nontransparent surface representation and displays only those exposed on the surface.

The analysis shows that only a relatively small number (~14) of the 46 residues mutated in non-Hodgkin's lymphoma are located on the surface of the complex (Fig. 3a, b). We find this to be the case for the residues mutated in colorectal or endometrial cancers (Supplementary Fig. 2c, d). Analysis of a higher resolution ATM model (PBD 6K9L, 4.27 Å) published subsequently[22] confirms that the majority of residues mutated in cancer are buried internally (Supplementary Fig. 3). We performed the same analysis on the ATR residues mutated in endometrial, colorectal, or breast cancer and also found that only a few residues are on the surface (Supplementary Fig. 2b). We infer that majority of conserved ATM/ATR residues mutated in cancer are buried within the respective enzyme complexes, irrespective of the cancer type.

**Solvent accessible ATM/ATR residues implicated in cancer**. We wished to examine whether the ATM and ATR residues mutated in a specific cancer type might localize to a distinct region(s) of the respective enzyme complexes. Unfortunately, we found the analysis challenging due to the large number of residues involved (e.g., Fig. 3c, d). As a means to mitigate this, we decided to focus on the few residues that are visible on the surface. In general, the solvent accessible residues do not cluster, irrespective of whether they are mutated in the same or different cancer types (Fig. 3c, d, f).

We performed a more detailed analysis on the TRD3 of ATM (Fig. 3c). TRD3 is a FAT domain motif (residues 2195–2475) with a long helical hairpin that interacts with the catalytic pocket of the other protomer and keeps the enzyme complex in a minimally active state (Fig. 3c, d)[9]. The model shows that four of the seven residues mutated in non-Hodgkin's lymphoma (D2441, D2448, L2452, and F2462) decorate one of the two helices of the hairpin, while four of the five residues mutated in endometrial cancer (L2345, A2354, R2400, and S2408) decorate the other (Fig. 3e and Supplementary Fig. 3c). Some of these residues are on the surface suggesting that they may facilitate binding of a protein partner(s). We find that the four solvent accessible residues mutated in endometrial cancers cluster and separated by <24 Å (Fig. 3e). In comparison, the closest solvent accessible residues mutated in colon cancer (W2300) and non-Hodgkin's lymphoma (D2448) are >30 Å apart (Fig. 3e). As a reference point, we examined the ATM residues involved in interacting with p53 and several other binding partners (residues 92–97)[9,23]. We find that these residues are solvent accessible and together cluster in an area that is ~13 Å at the widest (Fig. 3a, b).

**Only few ATM/ATR residues mutated in cancer localize to the active site**. The ATM/ATR kinase domain is ~400 residues long (i.e., ~15% of the polypeptide) and comprises the N- and C-lobes (Fig. 4a, b). The C-lobe harbors the critical activation and catalytic loops, as well as several regulatory motifs, including the PRD and FATC (Fig. 4a, b). We find that 42% and 32% of the conserved ATM and ATR residues mutated in cancer are in the kinase domains, representing approximately a threefold and twofold enrichment, respectively (Fig. 2b and Supplementary Fig. 2a). Notably, majority of these residues are found some distance away from the active site, suggesting that mutations at these residues modulate kinase activities via altered regulation instead of the catalysis per se (Fig. 4d, e and Supplementary Fig. 4a, b).

The mutated residues in the kinase domain are also cancer type specific (Fig. 4c). For instance, seven of the nine breast cancer-specific ATR residues are in the PRD (Fig. 4d). In contrast, none

of the ATR residues mutated in colorectal or endometrial residues is in the PRD (Fig. 4d, Supplementary Fig. 4a, and Supplementary Data 4). The PRD of ATM/ATR proteins normally functions to block substrate accessibility to the catalytic pocket (e.g., Fig. 4b). Upon activation, it moves away from the active site enabling the substrate access (e.g., Fig. 4a)[10,24]. Mutations in the PRD impairs TopBP1-dependent activation of ATR[24] and increased TopBP1 abundance has been linked to progression of hereditary breast cancer[25]. Together, these findings suggest that the PRD–TopBP1 interaction might be particularly relevant for the breast cancer development.

As in the case for the rest of the enzyme complex (Fig. 3 and Supplementary Fig. 2b, c), only a small number of the mutated residues in the kinase domain are solvent accessible; ~8/42 for ATM and ~8/22 for ATR (Figs. 3a–d and 4f–h, and Supplementary Fig. 4f, g).

**Identification of conserved Mec1 residues responsible for mediating the DDR**. As an independent means to address the genotype–phenotype relationship of ATM/ATR, we performed an unbiased genetic screen of *MEC1* to identify residues critical for the DDR. Budding yeast *S. cerevisiae* encodes two ATM/ATR proteins, Mec1 and Tel1. Mec1 shares ~24% identity with the human ATR and ~20% with the ATM; in comparison, Tel1 shares ~20% identity with the ATR and ~22% with the ATM (Supplementary Fig. 1b). Mec1, like ATR, is essential for viability and requires a constitutive binding partner Ddc2/Lcd1 for its function[26–30]. On the other hand, *tel1Δ*, unlike *atmΔ*, does not confer a notable growth defect or sensitivity to genotoxic stress, e.g., refs. [31,32]. Mec1 performs most functions of both ATM and ATR, and is widely regarded as the yeast functional ortholog of both ATR and ATM.

Briefly, the entire open reading frame of *MEC1* (7104 base pairs) was subjected to a random chemical mutagenesis and the mutagenized pool was screened for alleles that conferred sensitivity to hydroxyurea (HU) and/or methyl methanesulfonate (MMS), widely utilized replication stress- and DNA damage-inducing agents, respectively (Supplementary Fig. 5a). Importantly, the screen was performed in the absence of any second site suppressor mutation, for example, a deletion allele of *SML1*, an inhibitor of the ribonucleotide reductase[33], to select against *mec1* alleles that might impair its essential function(s). Analysis of ~10,000 viable *mec1* strains led to isolation of 15 missense alleles, each carrying a unique single amino acid alteration in the polypeptide (Fig. 5a, Supplementary Fig. 5a, Supplementary Data 5, and Supplementary Table 2).

Majority (9/15) of the mutated Mec1 residues are conserved in the human ATM and/or ATR. Moreover, six correspond to a residue mutated in cancer (Fig. 5a, c and Supplementary Data 6). Each of the 15 *mec1* alleles was transformed into a fresh wild-type (WT) strain background to confirm that the HU/MMS sensitivity is solely attributable to the mutation (Fig. 5d).

**Differential impact of *mec1* mutations on HU- versus MMS-dependent Rad53 activation**. Rad53 is the budding yeast CHEK2 effector kinase of the DDR, whose Mec1-dependent activation is required for resistance to HU and MMS[34,35]. In some mutants (e.g., *G1804N* and *C1985Y*), the activation signals are reduced to a level comparable to a lethal kinase dead (*mec1-kd*) mutant kept viable by a second site suppressor mutation *sml1Δ* (ref. [33]; Fig. 5b and Supplementary Fig. 5d). In others (e.g., *S1413F* and *G1546S*), the extent of Rad53 activation is comparable to the WT (Fig. 5b), suggesting that Rad53 activation is necessary but not sufficient for the resistance. These mutants might be impaired in activating other critical mediators of the DDR (e.g., Mrc1, RPA, and Rad9)[36,37]. It is

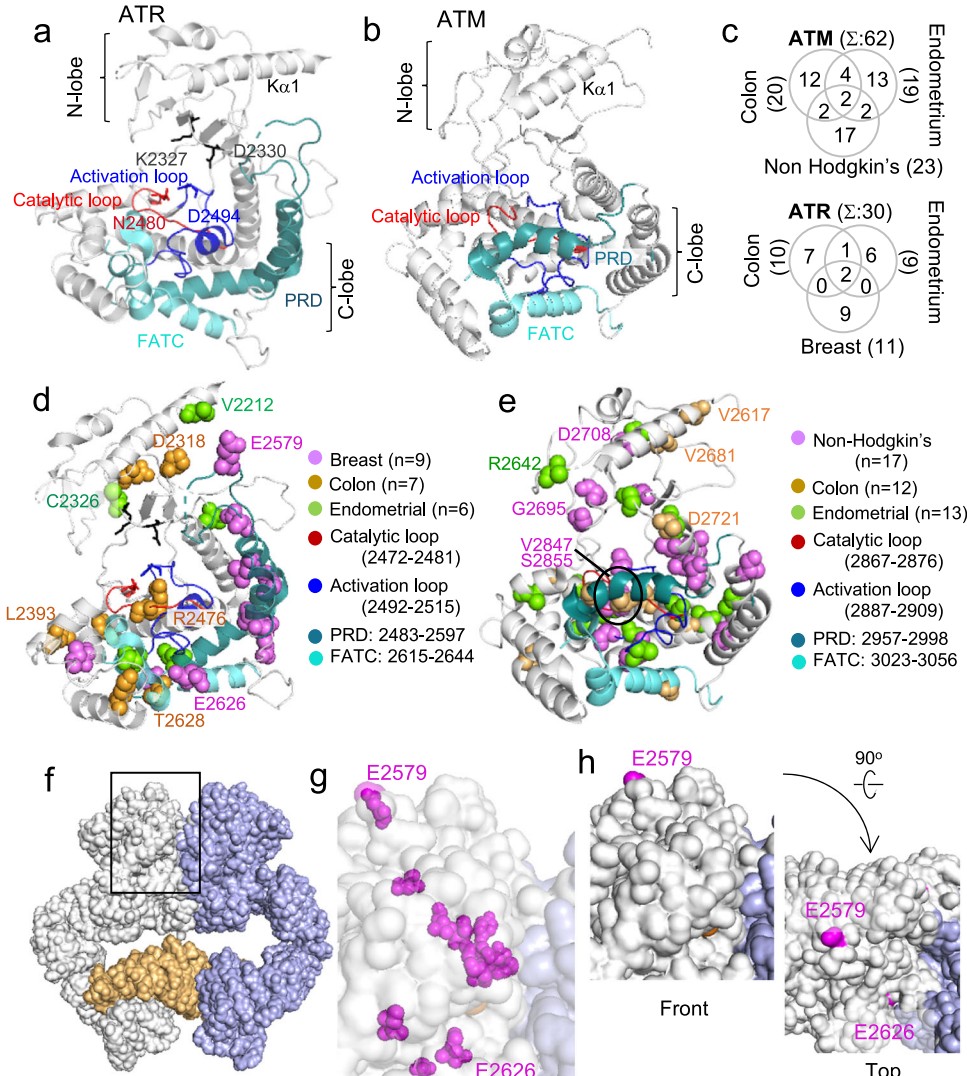

**Fig. 4 Molecular modeling analysis of ATM/ATR kinase domain residues implicated in cancer. a** Kinase domain of an active ATR enzyme complex (PDB 5YZO, 4.70 Å)[10]. The PRD is positioned away from catalytic center to allow substrate access. The K2327 and D2330 support ATP association. The N2480 and D2494 stabilizes $Mg^{2+}$ for catalysis[10]. **b** Kinase domain of an inactive ATM enzyme complex (PDB 5NPO, 5.70 Å)[9]. The PRD sterically hinders substrate access to the catalytic center. **c** Extent of overlap among the ATM/ATR residues mutated in the indicated cancer type that are the kinase domain (Fig. 2b, Supplementary Fig. 2a, and Supplementary Data 3 and 4). **d, e** The kinase domains of ATR (**d**) and ATM (**e**), showing location of the conserved residues mutated in the indicated cancer type. Labeled residues are solvent accessible (**h, g**) Supplementary Fig. 4f, g). **f** A surface representation of the ATR enzyme complex (PDB 5YZO, 4.70 Å)[10]. The left and right ATR polypeptides are white and blue, respectively, and Ddc2 is in bright orange. Black rectangle: the kinase domain regions highlighted in **g** and **h**. **g** Higher resolution image of the area highlighted by a black rectangle in **f**. Magenta spheres: residues mutated in breast cancer shown in **d**. The transparency setting was set to "on" to visualize both the buried and exposed residues. Only the E2579 and E2626 are exposed (**h**). **h** Same as in **g** except that the transparency setting was "off". Front and top views show that only the E2579 and E2626 are solvent accessible.

---

also possible that they might be impaired in an important catalysis-independent function(s)[38,39].

In some mutants, the extent of HU- and MMS-dependent Rad53 phosphorylation is comparable; for example, in a *mec1-S1413F* background, signals for both phosphorylation events are near normal (Fig. 5b), while in a *mec1-C1985Y* background, both HU and MMS-dependent activation is abolished (Fig. 5b). In others, the effects are stress specific; for example, in a *mec1-E2130K* background, the deficit is notably greater for HU but in a *G2779K* background, the effects are greater for MMS (Fig. 5b). The latter suggests that Rad53 activation in response to HU and MMS proceeds via genetically separable pathways, and that each pathway may depend on a different Mec1 residue(s).

**Critical Mec1 residues map to diverse locations in the enzyme complex.** We utilized a cryo-EM model structure of the Mec1–Ddc2 enzyme complex (PDB 5X60, 3.9 Å)[11] to visualize the 15 mutated residues (Fig. 6a). Reminiscent of the conserved ATM/ATR residues mutated in cancer (Fig. 3a–d, and Supplementary Figs. 2b–d and 3), majority (12/15) of the Mec1 residues are buried inside the enzyme complex (Supplementary Fig. 6b). To test whether these residues may have a structural role, we examined the impact of each *mec1* allele on resistance to elevated temperature, a widely utilized means of assessing structural integrity of a protein or complex. Seven of the 15 *mec1* mutants exhibited temperature sensitivity (Fig. 5d). The mutated residues in these alleles are found at diverse locations (Fig. 5c); the C467 at

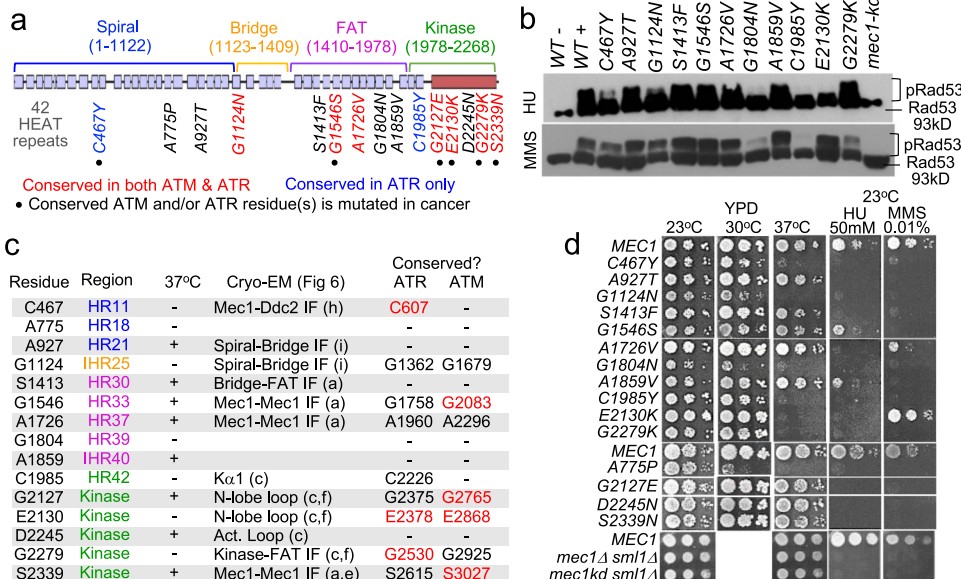

**Fig. 5 An unbiased genetic screen identifies conserved Mec1 residues required for the DDR. a** Mec1 comprises 42 HEAT repeats and a kinase domain[45]. An unbiased genetic screen identified 15 residues critical for mediating resistance to HU and/or MMS (Supplementary Fig. 5a, Supplementary Table 2, and Supplementary Data 5). Red: residue is conserved in both ATM and ATR. Blue: conserved in only ATR. ● The conserved ATM and/or ATR residue is mutated in cancer (**c**; Supplementary Data 3, 4, and 6; http://cbioportal.org). **b** Impact of the indicated *mec1* allele on HU- and MMS-dependent Rad53 activation. Western blot analysis was performed using a Rad53 antibody that detects both unphosphorylated and phosphorylated species (EL7.E1)[35]. "WT−": untreated negative control sample. "WT+": positive control sample. "*mec1-kd*": a kinase dead mutant in a *sml1Δ* suppressor mutation background used as a control. Locations of the unphosphorylated (93kD) and phosphorylated Rad53 species are indicated (Supplementary Fig. 5d). **c** Summary of the results presented in **b** and **d**, Fig. 6 and Supplementary Fig. 5c. Region: the colors correspond to the four domains in **a**. HR *HEAT Repeat* unit. IHR *Inter-HR*; the loop region following the indicated HR unit. 37 °C: growth at 37 °C (**d**). Cryo-EM (Fig. 6): location of the indicated residue in the Mec1–Ddc2 enzyme complex. The letters in the parenthesis corresponds to the relevant panel in Fig. 6. IF interface. Conserved?: the ATR/ATM residue corresponding to the indicated Mec1 residue (Supplementary Data 6). -: not conserved. Residues written in red are mutated in cancer (Supplementary Data 3, 4, and 6; http://cbioportal.org). **d** Impact of temperature, HU, and MMS on the indicated *mec1* strains ("Methods"). The *mec1Δ* and *mec1-kd* control strains are in a *sml1Δ* suppressor mutation background necessary to maintain viability; all other *mec1* mutants are in a *SML1* background.

the Mec1 and Ddc2 interface in the spiral domain (Fig. 6a, d, h), the A775 in the middle of the spiral domain (Fig. 6a), the G1124 at the interface between the spiral and bridge domains (Fig. 6a, d, i), and the N1804 in the FAT domain (Fig. 6a). Three temperature sensitive alleles carry a mutation in the kinase domain (Fig. 5c, d); the C1985 in the Kα1 (Fig. 6a, c), the E2130 in the second loop of the N-lobe (Fig. 6a–c), and the G2279 at the bottom of the kinase domain juxtaposed to a conserved HR unit in the FAT domain (Fig. 6a, d, f and Supplementary Fig. 7b).

Six (40%) of the 15 critical Mec1 residues are in the kinase domain, representing ~3-fold enrichment. Among the six, only the D2245 is in the active site (Fig. 6c and Supplementary Fig. 6a). The *D2245N* mutation abolishes both the HU- and MMS-dependent Rad53 phosphorylation, but not the resistance to heat (Fig. 5d and Supplementary Fig. 5c). This phenotype is consistent with a direct role of the D2245 in the enzyme catalysis. The remaining five kinase domain residues are more than ~20 Å away from the active site (Supplementary Fig. 6a), suggesting that mutations at these residues modulate kinase activity via altered regulation rather than the catalysis per se. The *G2127E*, *E2130K*, *G2279*, and *S2339N* alleles confer differential impact on HU- versus MMS-dependent Rad53 activation (Fig. 5b and Supplementary Fig. 5c). It is possible that the mutated residues play a role in some aspect(s) of a stress-specific Mec1 function.

Previously, Longhese and colleagues isolated two separation of function alleles, *mec1-100* and *mec1-101*, which do not comprise Mec1's essential function(s) but impair the DDR[33]. The *mec1-100* carries two mutations, the F1179S and N1700S; the *mec1-101* carries three, the V225G, S552P, and L781S. Notably, the five mutated residues are nearby the residues identified in current

study; for example, the S552 mutated in the *mec1-101* is just ~6.5 Å from the C467 (Supplementary Fig. 6d) and the N1700 mutated in the *mec1-100* is in the middle of the five residues identified in our screen (Fig. 6f). Proximity of the critical residues identified in two independent studies provide further support for the likely functional importance of these regions.

**Structural similarities among the conserved Mec1, ATR, and ATM residues.** Nine of the 15 Mec1 residues critical for the DDR are conserved in ATR and/or ATM (Fig. 5a). Among the conserved, four ATM (G2083, G2765, E2868, and S3027) and three ATR (C607, E2378, and G2530) residues are mutated in cancer (Fig. 5a, c). Molecular modeling analysis shows that these conserved residues are found at the corresponding locations in the respective enzyme complexes, suggesting that their functions are also likely to be conserved (Fig. 6 e, i, j and Supplementary Fig. 7).

The ATR G2375, mutated in endometrial cancer, corresponds to the Mec1 G2279 (Fig. 5c and Supplementary Data 6). Both residues, as well as the corresponding ATM G2925, are at the bottom of the respective kinase domains, next to a HR unit that is located ~800 residues away (Fig. 6h and Supplementary Fig. 7b). We find that these HR units contain four residues that are conserved across Mec1, ATR, ATM, and Tel1 (Fig. 6g). Remarkably, two of the four ATR (D1687 and G1691) and two of the four ATM (L2005 and D2016) conserved residues are mutated in cancer (Supplementary Data 3, 4, and 6). The cancer-associated residues are next to the kinase domain residues ATR G2375 and ATM G2925, respectively, providing further support for the functional relevance of the FAT kinase domain interface

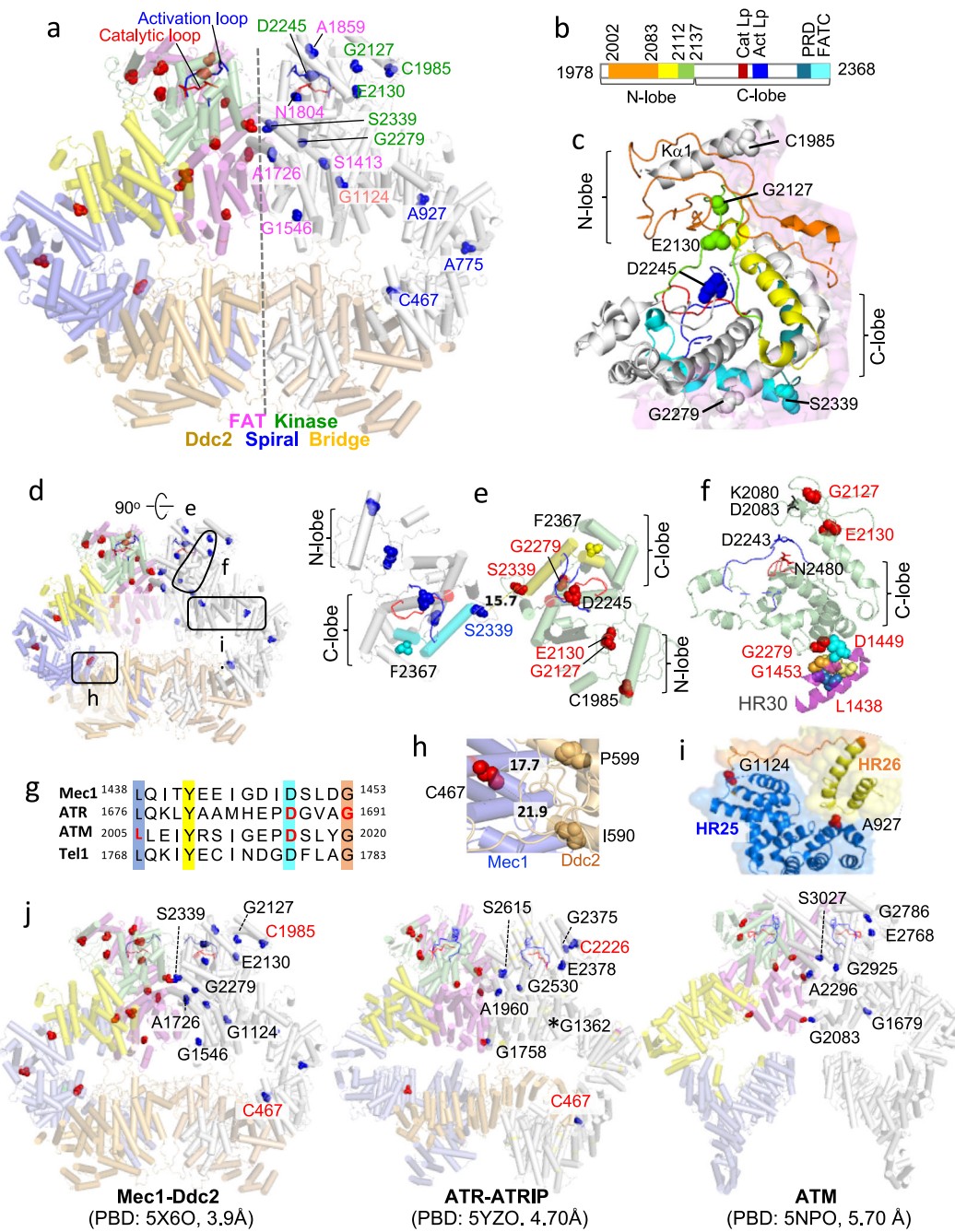

int the ATM, ATR, and Mec1 enzyme complexes (Fig. 6f and Supplementary Fig. 7b).

## Discussion

We addressed the elusive genotype–phenotype and structure–function relationships of ATM/ATR proteins by examining the conserved ATM/ATR residues mutated in cancer and performing an unbiased functional genetic analysis of *MEC1*. Our analyses unveil that only a few of the ATM/ATR residues implicated in cancer are in the active site of the respective enzyme complex, suggesting that loss of the intrinsic kinase activity is infrequent in carcinogenesis (Fig. 4c, d and Supplementary Fig. 4a, b). This would make sense from a cancer cell perspective because the loss of kinase activity would impact a range of processes that ATM/ATR proteins govern, including cellular proliferation; accordingly, such a mutation would confer little selective

advantage. In further support for this notion, we find that only 1 of the 15 nonlethal *mec1* alleles impaired in the DDR carries a mutation in the active site (Supplementary Fig. 6a, D2245N).

The majority of the ATM/ATR residues implicated in cancer are buried inside the respective enzyme complexes (e.g., Fig. 3). These residues may have a structural role, which could account for the findings that a large number of the cancer-associated ATM mutations destabilize the gene product[13]. Findings of our *MEC1* analysis also support this view: majority (12/15) of the Mec1 residues critical for the DDR are buried inside the enzyme complex (Supplementary Fig. 6b) and seven *mec1* alleles confer temperature sensitivity, a widely utilized readout for structural deficits (Fig. 5d).

All 15 *mec1* mutants are viable and proficient in unchallenged proliferation. This suggests that each mutant complex, including those that confer temperature sensitivity, is structurally sound under unchallenged conditions and maintains the intrinsic kinase

**Fig. 6 Molecular modeling analysis of Mec1 residues involved in the DDR. a** The 15 critical Mec1 residues are mapped onto a cryo-EM model of the inactive dimeric Mec1–Ddc2 enzyme complex (PDB 5X6O, 3.9 Å)[11]. The dashed black line in the middle denotes the twofold symmetry axis. The complex is in cylindrical helices representation. The protomer on the left-hand side is in blue, yellow, violet, and green, representing the spiral, bridge, FAT, and kinase domains, respectively. The protomer on the right-hand side is in white. Both Ddc2 chains are shown in light brown. The 15 residues in the left and right protomers are shown in red and blue sphere representation. **b** Schematic representation of the Mec1 kinase domain, which comprises the N- and C-lobes. The N-lobe contains two extended loops shown in orange (residues 2002–2083) and green (2112–2137). The first loop is preceded by the kα1 (1978–2002) shown in white (**c**). Between the two loops is another helical (2083–2112 shown) shown in yellow (**c**). The C-lobe contains the critical activation and catalytic loops, and several regulatory motifs including the PRD and FATC. **c** A round helices representation of the Mec1 kinase domain. The six residues identified in our screen are shown in sphere representation in the colors corresponding to those shown in **b**. The neighboring FAT domain is shown in faint pink. **d** Lower resolution image of the model in **a**, showing the areas that are highlighted in **e**, **f**, **h**, and **i**. **e** Top view of the dimeric enzyme complex in cylindrical helices representation depicting proximity of the two S2339 residues; only the two kinase domains are shown. The kinase domain on the left is in white except the FATC, which is in cyan. The kinase domain on the right is in pale green except the FATC, which is in yellow. The activation and catalytic loops are shown in blue and red, respectively. The six mutated residues are shown in blue (left) or red (right). The two S2339 residues are separated by 15.7 Å. Red label: the corresponding ATM and/or ATR residue is mutated in cancer. Cyan and yellow: FATC. F2367: N-terminus of the Mec1 polypeptide in the model; the last residue, W2368, is missing in the published Mec1–Ddc2 complex[11]. **f** The G2279 is at an interface between the kinase domain and the HR30 of the FAT domain. The HR30 is ~800 residues away from the G2279 and contains four residues that are conserved in ATR, ATM, and Tel1 (**g**). Red residue: corresponds to an ATM and/or ATR residue mutated in cancer. The K2080 and D2083 correspond to the ATR K2327 and D2330 involved in ATP association, respectively. The N2229 and D2243 correspond to the ATR N2480 and D2494 that stabilize $Mg^{2+}$ for catalysis, respectively[10] (Fig. 4a). The G2127 and E2130 are two other kinase domain residues identified in the current study (**a**, **c**). **g** Sequence alignment of the Mec1 HR30 and the corresponding HRs in ATR, ATM, and Tel1. The four shaded residues are conserved across the four proteins. Color of the shade correspond to the color of each residue in **f**. The ATM L2005 and D2016, and the ATR D1687 and G1691 are mutated in cancer (Supplementary Fig. 7b, and Supplementary Data 3, 4, and 6). **h** The Mec1 C467 is at the interface between the Mec1 spiral domain (blue) and Ddc2 (light brown). The C467 is ~20 Å from the Ddc2 I599 and P599, which correspond to the ATRIP L662 and P671 mutated in cancer, respectively (http://cbioportal.org). **i** The Mec1 A927 in the spiral domain and the G1124 is in the linker (1122–1148)[11] are found at the interface between the spiral (blue) and bridge (yellow) domains. **j** Cryo-EM models Mec1, ATR, and ATM enzyme complexes showing location of the conserved residues critical for resistance to HU/MMS in yeast (Fig. 5a, c). The seven black residues are conserved in Mec1, ATR, and ATM. The two red residues are conserved only in Mec1 and ATR. *ATR G1362 is in a flexible region (Supplementary Fig. 7d) and missing in the published model[10].

activity necessary for its essential functions. It is possible that our mutants are hypomorphs with a diminished kinase activity, which might be sufficient for the essential function(s), but not for the DDR, which may require a higher level of Mec1 activity. It is also possible that they are separation of function mutants impaired only in a HU/MMS-dependent function(s), for example, in interacting with a specific HU and/or MMS-dependent binding partner(s). Notably, since the majority of the mutated residues are buried inside, such an effect might be indirect; alternatively, some of the buried residues might become exposed following enzyme activation.

Evidence suggests that ATM/ATR activation entails a structural change that is triggered by the binding of a stress-specific activator, such as Nbs1 for ATM and Dpb11 for Mec1 (refs. [36,40]). During the activation process, some regions of the enzyme complex might experience a localized steric and/or mechanical stress (Fig. 7a). It is likely that under such conditions, the enzyme activation would depend on relief of the stress. Some of the Mec1 residues identified in the current study might play a role (Fig. 7a). Mutations at such a residue, which we will refer to as an "intermediary residue", would hinder stress-dependent activation by blocking the conformational change (Fig. 7b) or causing the enzyme complex to come apart having succumbed to the stress (Fig. 7c). Notably, these mutations would not impact the essential function(s) of the enzyme complex (Fig. 7).

While best understood for their DDR functions, ATM/ATR proteins are also known to be involved in a number of tissue specific and/or developmentally controlled processes. For example, in response to the developmentally programmed meiotic DNA double-strand breaks (DSBs), Mec1 and Tel1 activate meiotic chromosomal proteins Hop1 and Rec114 instead of Rad53 (refs. [31,41]). Activation of Hop1 ensures the essential inter-homolog bias in meiotic DSB repair, while activation of Rec114 prevents inappropriate DSB catalysis. Similarly, ATM/ATR activate the mammalian counterparts, HORMAD1, 2 and REC114, respectively, in response to meiotic DSBs; moreover, the latter

two are implicated the gonadal dysgenesis in A-T patients[42,43]. Combining these with the current findings that the ATM and ATR residue mutated in cancer are tissue specific (Fig. 2b, c and Supplementary Fig. 2a), and that some regions of the respective enzyme complexes are enriched for residues that are mutated in a specific cancer type (Fig. 3c–e, Fig. 4d and Supplementary Fig. 4a) suggest the cancer relevance of separation of function ATM/ATR mutations leading to the loss of a tissue-specific function(s).

## Methods

**In silico analysis of ATM/ATR.** Somatic mutations of ATM/ATR were collected from 46,588 tumor samples in the cBioPortal database (http://cbioportal.org)[15,16]. Each of the 921 ATM and 746 ATR residues mutated to a different residue(s) in cancer was manually examined to determine whether it is conserved in Mec1[ATR] and/or Tel1[ATM]. To determine the extent of conservation between the human ATM/ATR and their homologs, sequences of the ATM/ATR proteins from *Homo sapiens*, *M. musculus*, *X. laevis*, *A. thaliana*, and *S. cerevisae* were downloaded from the NCBI website and saved as FASTA files. Multiple sequence alignment was performed using Clustal-Omega using the FASTA files as the input using default settings.

**Molecular modeling analysis.** Cryo-EM model structures of the human ATR-ATRIP dimer (PDB: 5YZ0) and ATM dimer (PDB:5NP0) were downloaded from the Protein Data Bank website http://www.rcsb.org/pdb/. Coordinates for the Mec1[ATR]–Ddc2[ATRIP] dimer complex was provided by Dr. Gang Cai (University of Science and Technology of China, Hefei, China). All structure images were generated using PyMol (The PyMOL Molecular Graphics System, Version 2.0 Schrödinger, LLC).

### Yeast manipulations

*Strains and media*. All strains utilized in current study are isogenic derivatives of RCY18 (*MAT a ho::LYS2, lys2, ura3, leu2::hisG, ade2::LK, his4x, mec1[ATR]Δ::LEU2* plus *pARS/CEN/URA3-MEC1 [ATR]*). Mutant alleles of *mec1 [ATR]* were generated by hydroxylamine random mutagenesis[44] of an ARS/CEN/ADE2 plasmid carrying the entire *MEC1 [ATR]* ORF. The mutagenized pool was transformed into RCY18 and the resultant *ADE* prototrophic strains were replica plated onto to 5-fluoroorotic acid (5FOA) plates to "shuffle out" the *URA3-MEC1 [ATR]* plasmid (Supplementary Fig. 2a). Approximately 10,000 5FOA resistant colonies were screened for sensitivity to HU and/or MMS, which ultimately led to isolation of the 15 alleles utilized in the current study (Fig. 2a, Supplementary Fig. 5a, and Supplementary Table 2). During the screen, yeast cells were grown in synthetic media (2% glucose and

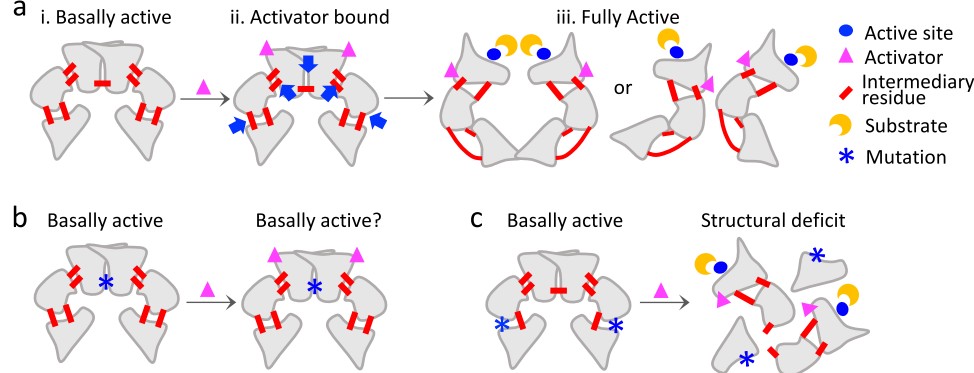

**Fig. 7 Model for stress-dependent activation of ATM/ATR. a** Genotoxic stress-dependent binding of an activator triggers the conformational change from a basally active dimeric enzyme complex to a fully active dimeric or monomeric enzyme. (i) The enzyme complex under unchallenged condition performs normal cellular functions. Substrate access to the active site (blue circle) is restricted to prevent inappropriate activation of the DDR. (ii) Stress induces binding of an activator (magenta triangle), which triggers the conformation transition. Some areas of the complex may experience localized strain (blue arrows). (iii) Fully active dimeric or monomeric enzyme. Intermediary residues at various locations (red bars) facilitate the enzyme activation by providing steric/mechanical flexibility. **b**, **c** Two possible ways by which mutations at an intermediary residue might impair the DDR. **b** The mutation blocks the conformational transition. **c** The mutant complex comes apart during the transition. Note that in both cases, the mutant enzyme complex would be proficient in performing its basic function(s).

0.67% yeast nitrogen base plus ammonium sulfate) supplemented appropriate dropout amino acid mix. After the screen, yeast cells were grown in rich growth medium YPD (1% yeast extract, 2% bacto peptone, and 2% glucose). All yeast medium reagents were purchased from Formedium (Norfolk, UK).

*Stress sensitivity test*. Strains were grown up from −80 °C 20% (v/v) glycerol stock, first on YPG plates (1% yeast extract, 2% bacto peptone, 2% glycerol, and 2% agar) to select against petite mutants (mutants lacking mitochondria). After 1–2 days growth, the cells were streaked for single colonies on YPD plates and incubated further for 2–3 days. Colonies were used to inoculate 2 ml YPD medium and grown overnight at 25 °C. For each strain, the appropriate volume of overnight culture was added to 1 ml ddH$_2$O to give an optical density (OD$_{600\,nm}$) of 0.5. A 1/10 serial dilution with water was made. The strains were then transferred onto the YPD plates supplemented with either 50 mM HU or 0.02% MMS using RoToR HDA (Singer Instrument, Somerset, UK). The plates were allowed to dry and then incubated at 25 °C for 2–3 days before imaging. For temperature sensitivity, the plates were incubated at 37 °C.

**Western blot analysis**. Western blot analysis was performed on 20% TCA (tri-chloroacetic acid) extracts prepared from culture volumes corresponding to 10 units of OD600. Status of Rad53 activation was assessed using the EL7.E1 monoclonal antibody at 1:20 dilution of the aliquot prepared from Dr. Marco Foiani's laboratory (gift from M. Foiani; IFOM, Milan, Italy). Secondary antibody used was goat α-mouse IgG (Abcam [ab97040], 1:10,000).

**Statistics and reproducibility**. Three or more biological replicates of each strains were analyzed for all Western blot analysis and HU/MMS/heat sensitivity tests. The results were highly reproducible and the data presented in the manuscript are representative.

**Reporting summary**. Further information on research design is available in the Nature Research Reporting Summary linked to this article.

## Data availability
The datasets generated during and/or analyzed during the current study and the data that support the findings of this study are available from the corresponding author on reasonable request.

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

## Acknowledgements

The authors thank Katrin Rittinger and Steve Smerdon for insightful discussions on molecular modeling, Gang Cai for providing the coordinates for dimeric Mec1–Ddc2 complex, and Chris Staple and Edgar Hartsuiker for critical reading of the manuscript. This work was supported Bangor University PhD studentship award to E.W. and by grants from North West Cancer Research to R.S.C. (CR961 and CR1161).

## Author contributions

Conceptualization: R.S.C. Formal analysis: R.S.C. and E.W. Investigation: R.S.C., I.C.-S, M.V., and E.W. Resources: R.S.C. and E.W. Writing-original draft preparation: R.S.C. and E.W. Visualization: R.S.C. and E.W. Supervision, project administration, and funding acquisition: R.S.C.

## Competing interests

The authors declare no competing interests.
