## [Peer Review File · Communications Biology]

Reviewers' comments:

Reviewer #1 (Remarks to the Author):

This is a very nice study that identifies cancer associated mutations in ATM and ATR, determines whether the mutated amino acids are conserved through evolution and maps them onto the structures of the proteins. The results could be helpful for determining the physiological consequences of cancer associated mutations in ATM and ATR for precision medicine.

On lines 87-95 could the authors please clarify what they mean by the cancers with a higher ATM or ATR mutations frequency have more mutated residues. Does the number of depend on the number of samples in the cancer type and has a corrections been made to account for this?

Also, isn't this an obvious point? If a cancer has a higher ATM mutation frequency the number of mutations in ATM is more? Or are you saying that there are no mutation hotspots so the total number of residues that are mutated increases rather than the number of mutations.

Alternatively perhaps are you referring the frequency of mutations in other genes in ATM deficient cancers (ie loss of ATM confers genome instability). Please clarify.

Line 97: of the above (add the)

Line 100: please list the accession numbers of the ATM/ARR sequences used. A table in the supplementary would be adequate.

Line 286: was manually examined (change is to was)

Line 108: on a few (add a)

Reviewer #2 (Remarks to the Author):

Cancer genome datamining and functional genetic analysis implicate multiple mechanisms of ATM/ATR dysfunction underpinning carinogenesis
Erik Waskiewicz, Michalis Vasiliou, Isaac Corcoles-Saez and Rita S. Cha

Waskiewicz et al. present an analysis and experimental design used to identify important ATM, ATR, and Mec1 amino acid residues important for function and are cancer associated. Here, the authors utilize the cBioPortal to map cancer associated ATM/ATR missense-mutations to the position in the polypeptide sequence and already available three-dimensional cryo-EM structures of ATM/ATR. Several of these resides mutated in cancer appear to be cancer type specific. They take this further and map these corresponding residues onto the yeast homolog, Mec1. Additionally, they find that most of these residues are localized internally in the structure, with few localizing to the active site or being solvent exposed. They next performed a mutational screen of the Mec1 and found a similar pattern of missense-mutations being found along the entire length of the protein, with most found internally and 14 residues conserved in both ATM/ATR. They tested several mutants they identified and show that there is impaired activation of the downstream Mec1 target, Rad53.

Although ATM/ATR cancer associated mutations have been studied before, the analysis presented here is more complete, and the mutants identified from the Mec1 mutagenesis screen can be useful to guide further studies. Additionally, the study here complements their meta-analysis with a genomic

screen to show that some of these mutants in cancer directly affect function, at least in the yeast Mec1 homolog. I have the following comments:

- (line 108) "The results show that the location is cancer type specific (Fig 2A)" – This statement is referencing Fig 2B, not 2A.
- (line 113-116) "...only two of the 88 ATR residues are common to the three cancer types" – although the mutations themselves exhibit cancer type specificity, are these mutations in the same domain?
- (lines 125-127) "relatively small number of..." and "few of the ATMP residues..." – for clarity, state the actual number of residues
- (lines 173-177) "Analysis of ~10 000 viable mec1ATR strains led to isolation of 14 missense alleles ... found along the entire length of the polypeptide" The authors previously state that cancer associated mutations are not enriched in the kinase domain. Although the 14 mutations found are indeed across the entire length, 6/14 of these residues are in the kinase domain, and 4/7 the conserved cancer residues are in the kinase domain. This might suggest that the more relevant mutants that confer sensitivity to HU and MMS are in the kinase domain, at least in the yeast Mec1.
- (lines 193-194) "...might be involved in interacting with a substrate or binding partner" – is there any insight as to which interactions may be affected that can be mentioned? For example, are there a decreased interaction among Mec1 and its activators (i.e. Ddc2, Dpb11) among these mutants?
- (lines 202-204) "...exhibit notable temperature sensitivity, suggestive of a structural role..." though this tests for the stability of the protein, it would also be informative to see how the level of proteins actually vary at 30oC. Thus, Mec1 protein levels could be tested to see altered protein levels, increased degradation, etc.
- (line 212) As an alternative interpretation, perhaps replication stress resistance requires a higher level of Mec1 activity than does the essential function(s). Basal activity is sufficient for viability, but not for stress resistance.
- (line 230) Studies of the separation of function mec1-100 allele are an exception and would be worth discussing.
- (line 234) Rad53 phosphorylation is still evident in most the mutants, and so the residues are not required for phosphorylation, but rather they promote full Rad53 phosphorylation.
- (Figure 4) "The kinase domains of ATM (B) and ATR (C)..." should be corrected to "The kinase domains of ATM (B) and ATR (D)"
- (Figure 5) Since the screen was performed to identify alleles sensitive in MMS or HU, and also since the western blot shows difference in Rad53 activation between MMS or HU among some alleles, it might be informative to display which mutants were identified to be sensitive in MMS and/or HU.
- (Figure 5h) It is difficult to tell which mutant label refers to which spots on the sensitivity assay. There appears to be two replicates for each, a black line to connect the two could be placed, with the mutant labels above it. Additionally, the MMS images appear pixelated and blurry. A higher resolution image should be provided.
- (Figure 5h) What is the genetic background of the MEC1 deletion strain used as a control in the sensitivity assay? Since it is essential, there must be an additional suppressor mutation.
- (Figure 5h) Do any of these mutant alleles exhibit a growth defect in just unperturbed/no drug? From the sensitivity assay, it appears not, but it is hard to tell. For example, C1985Y could exhibit some sickness, also considering that it is temperature sensitive, but it could also be due to uneven spotting of the culture.
- (Figure S2) Although this is out of the scope of the paper, how do the 24 / 50 mutations identified in the screen, that are not missense mutations, compare with the non-missense mutations identified in cancer-associated mutations of ATM/ATR? If they are similar or also conserved, this would further support Mec1 as a good model to understand ATM/ATR cancer mutations. In addition, it would be nice to see this available data, instead of only the missense mutation data. Together, all mutations identified would provide a nice data set for the literature.

Reviewer #3 (Remarks to the Author):

The manuscript by Waskiewicz et al. presents analysis of mutations in ATM and ATR, master protein kinases involved in DNA replication and repair. They present results from three different aspects: 1) datamining of the cancer database 2) correlating these mutants with available structural information and 3) yeast genetic screens. The datamining has generated some useful results. The comparisons on the number and locations of the ATM and ATR mutations found in different cancer types are interesting. However the manuscript does not go beyond that and falls short in exploring the potential causes or the implications of these differences. For the second part the authors intended to relate these mutations to structures. This could have provided some useful insights. However, the authors simply mapped these mutations onto available ATR/Mec1 and ATM/Tel1 structures, all of which were at modest resolution, not taking advantage of all the recent structural information, including some high resolution structures. Further, a number of studies have already analysed in details of the potential structural consequences of some of the key cancer mutations, often as part of structural studies, the authors fail to quote these studies or to relate their analysis with those published. In addition to ATR and ATM, a number of other highly related PIKKs proteins such as Tel1, mTOR and DNA-PKc, have been reported and many key functional regions have been discussed in detail. The current study lacks sufficient depths in analysing the potential structural consequences of these mutations apart from whether they are in the active site, inside or on the surface of the structure. The genetic screens of *mec1* identified some interesting mutations that render cells sensitive to HU/MMS agents which induces replication stress and DNA damage respectively. They measured levels of phosphorylated Rad53, as a measure of kinase activity, as well as cell viability. However, the results are presented sketchily, without detailed analysis of the relative activities and correlations. Furthermore, the results under HU and MMS challenges are not discussed separately nor the relationship between kinase activity and cell viability in details.

Overall the study is limited in scope and the manuscript presents limited new knowledge on ATR/ATM, some of the conclusions and statements in the manuscript are inaccurate or unjustified/supported. It would be important if the authors could conduct in-depth structure-function analysis with all available structural information of related PIKKs. Further, there are a significant amount of cancer mutations in ATM, and it would provide interesting and insightful knowledge if the authors carry out genetic analysis of Tel1 and/or conduct functional analysis of some of the interesting cancer mutations. Together, the new studies and analysis will provide insights into potential structural and functional consequences of these cancer mutations.

Specific comments and suggestions:

1. The authors chose mutations that were selected based on conservation between species – this seems appropriate. However, sequence alignments between these proteins become difficult towards the N-terminal regions, due to species-specific insertions and variable numbers of HEAT repeats. Whilst the authors describe in the methods that they performed alignments in Clustal Omega, do other alignment programs produce similar or different results? Does a structure-based sequence alignment also produce similar results? This is especially important as there are structures of both mammalian and yeast ATM and ATR, which would permit structure-based alignments to be made and therefore strengthen the comparisons made between yeast models and human disease mutations.
2. The unbiased genetic screen of *Mec1* is interesting and produces a number of mutations in the *MEC1* gene that reduce viability when challenged by genotoxic agents or heat shock. Importantly, the authors should demonstrate that each strain show similar expression levels of *Mec1* at the protein level. Clearly the *Mec1*-delta and the kinase-dead strains are viable and show similar sensitivity to

other mutants when challenged with MMS, HU, but not heat. This is an important control to confirm that the mutations isolated do not result in loss of protein due to destabilisation of the structure leading to proteasome-mediated degradation, for example. Additional consideration should be given to whether mutations alter nuclear localisation of the protein kinase, which would also produce phenotypes similar to those presented in the screen if not allowed to localise correctly.

3. Some of the conclusions/statements are inaccurate or unjustified. Below are a few (but nonexclusive) examples:

“only a small fraction of the critical residues is in the active site of the respective enzyme complexes, suggesting that loss of the intrinsic kinase activity is rare in carcinogenesis “

This sentence/conclusion is not justified/supported by the results presented in the manuscript.

“under unchallenged conditions, the dimeric complex exists as a minimally active enzyme, in which substrate-access to the active site is sterically hindered by several conserved features, including the PIKK-regulatory domain (PRD) in the kinase domain“

This is not strictly true for ATR/Mec1

Page 5 line 130: “is buried inside the enzyme complex, implicating a structural or regulatory role(s). “

This is not necessarily true as the only structures available are in autoinhibited state. These residues might be exposed or important in other functional states. Furthermore, the majority of residues within a protein are inside, thus the percentage of mutant residues are probably proportional to the percentage of surface residues within a structure and this observation is thus expected. Similarly, in terms of activation loop and catalytic loop, the size (< 10% of the kinase domain) is roughly in proportion to the relative number of mutations within these active site (~ 6 %). What is interesting is that mutations are disproportionally concentrated in the kinase domain (42% and 26%) which only occupies 10-15% sequence, suggesting many of these mutations are likely to modulate kinase activities, probably via altered regulation instead of catalytic activity per se.

Authors' response to Reviewers' comments

Please note that our responses are in **Arial font in blue** with actions highlighted in yellow.

Reviewer #1 (Remarks to the Author):

This is a very nice study that identifies cancer associated mutations in ATM and ATR, determines whether the mutated amino acids are conserved through evolution and maps them onto the structures of the proteins. The results could be helpful for determining the physiological consequences of cancer associated mutations in ATM and ATR for precision medicine.

On lines 87-95 could the authors please clarify what they mean by the cancers with a higher ATM or ATR mutations frequency have more mutated residues. Does the number of depend on the number of samples in the cancer type and has a corrections been made to account for this?

To clarify, we included the number of samples examined for each cancer-type in Fig 1E. This shows that there is no correlation between the sample size and the mutation frequency.

Also, isn't this an obvious point? If a cancer has a higher ATM mutation frequency the number of mutations in ATM is more? Or are you saying that there are no mutation hotspots so the total number of residues that are mutated increases rather than the number of mutations. Alternatively perhaps are you referring the frequency of mutations in other genes in ATM deficient cancers (ie loss of ATM confers genome instability). Please clarify.

The referee is correct to point out an expectation that there might be a correlation between mutation frequency and the number of mutated residues. **To convey this point, we added "As expected" in line 87.** However, there are exceptions as described in lines 90-92. We agree with the referee that this likely reflects the absence of any strong mutational hotspots. We explicitly state this in lines 92-94. All the analysis performed in the manuscript refers to mutations in only ATM/ATR.

Line 97: of the above (add the)

We added "the".

Line 100: please list the accession numbers of the ATM/ARR sequences used. A table in the supplementary would be adequate.

The information is provided in Fig S1

Line 286: was manually examined (change is to was)

We changed "is" to "was". Line 286 in our original manuscript is line 375 in the revised manuscript.

Line 108: on a few (add a)

We added "a".

Reviewer #2 (Remarks to the Author):

Cancer genome datamining and functional genetic analysis implicate multiple mechanisms of ATM/ATR dysfunction underpinning carcinogenesis

Erik Waskiewicz, Michalis Vasiliou, Isaac Corcoles-Saez and Rita S. Cha

Waskiewicz *et al.* present an analysis and experimental design used to identify important ATM, ATR, and Mec1 amino acid residues important for function and are cancer associated. Here, the authors utilize the cBioPortal to map cancer associated ATM/ATR missense-mutations to the position in the polypeptide sequence and already available three-dimensional cryo-EM structures of ATM/ATR. Several of these residues mutated in cancer appear to be cancer type specific. They take this further and map these corresponding residues onto the yeast homolog, Mec1. Additionally, they find that most of these residues are localized internally in the structure, with few localizing to the active site or being solvent exposed. They next performed a mutational screen of the Mec1 and found a similar pattern of missense-mutations being found along the entire length of the protein, with most found internally and 14 residues conserved in both ATM/ATR. They tested several mutants they identified and show that there is impaired activation of the downstream Mec1 target, Rad53.

Although ATM/ATR cancer associated mutations have been studied before, the analysis presented here is more complete, and the mutants identified from the Mec1 mutagenesis screen can be useful to guide further studies. Additionally, the study here complements their meta-analysis with a genomic screen to show that some of these mutants in cancer directly affect function, at least in the yeast Mec1 homolog. I have the following comments:

- (line 108) “The results show that the location is cancer type specific (Fig 2A)” – This statement is referencing Fig 2B, not 2A.
We changed “2A” to “2B”.
- (line 113-116) “...only two of the 88 ATR residues are common to the three cancer types” – although the mutations themselves exhibit cancer type specificity, are these mutations in the same domain?
We added Fig 3C-F, Fig 4D-G and Fig S4 showing location of the cancer type-specific ATM/ATR residues. We also discuss regions of the enzyme complexes that appear to be enriched for residues mutated in a specific cancer type (lines 154-162). For example, the enrichment of breast cancer associated residues in the PRD of ATR (lines 174-180; Fig 4D; Fig S4A).
- (lines 125-127) “relatively small number of...” and “few of the ATMP residues...” – for clarity, state the actual number of residues
We provide the numbers in lines 123-125.
- (lines 173-177) “Analysis of ~10 000 viable *mec1*ATR strains led to isolation of 14 missense alleles ... found along the entire length of the polypeptide” The authors previously state that cancer associated mutations are not enriched in the kinase domain. Although the 14 mutations found are indeed across the entire length, 6/14 of these residues are in the kinase domain, and 4/7 the conserved cancer residues are in the kinase domain. This might suggest that the more relevant mutants that confer sensitivity to HU and MMS are in the kinase domain, at least in the yeast Mec1.
We agree with the referee and amended the text accordingly in lines 246-247. We would also like to alert the reviewer of an error in the original manuscript; we examined 15 not 14 alleles. We apologize for the confusion. This error has been corrected throughout the revised text.
- (lines 193-194) “...might be involved in interacting with a substrate or binding partner” – is there any insight as to which interactions may be affected that can be mentioned? For example, are there a decreased interaction among Mec1 and its activators (i.e. Ddc2, Dpb11) among these mutants?
Currently, there is no antibodies against Mec1. We tested the published HA-MEC1 and MYC-MEC1 strains and found that they exhibit modest HU sensitivity, indicating that the tags impact protein function and/or stability. Given this, we are reluctant to utilize a tag because the results are likely to be

misleading. Mec1 antibody-production has been one of our top priorities for some time. We have raised and tested four polyclonal antibodies against two Mec1 peptides but none was found to be specific to Mec1. We plan to try again taking advantage of the available structural information. We will perform the experiment as soon as suitable antibodies become available.

- (lines 202-204) “...exhibit notable temperature sensitivity, suggestive of a structural role...” though this tests for the stability of the protein, it would also be informative to see how the level of proteins actually vary at 30oC. Thus, Mec1 protein levels could be tested to see altered protein levels, increased degradation, etc.

As mentioned above, there is no antibodies against Mec1 and epitope tags likely impact the protein function and/or stability. We plan to determine the impact of mutations on Mec1 abundance under unchallenged conditions and in response to different types of stresses as soon as a suitable antibody becomes available.

- (line 212) As an alternative interpretation, perhaps replication stress resistance requires a higher level of Mec1 activity than does the essential function(s). Basal activity is sufficient for viability, but not for stress resistance.

We agree and amended lines lines 344-5 accordingly.

- (line 230) Studies of the separation of function *mec1-100* allele are an exception and would be worth discussing.

We thank the reviewer for pointing out this highly relevant study. Remarkably, the residues mutated in *mec1-100* and *mec1-101* co-localize with the residues identified in our study, further signifying functional importance of the regions. We added a paragraph discussing this (lines 310-318) and images showing the co-localization (Fig 6F; Fig S6D).

- (line 234) Rad53 phosphorylation is still evident in most the mutants, and so the residues are not required for phosphorylation, but rather they promote full Rad53 phosphorylation.

We agree and amended the text (lines 215-21) accordingly.

- (Figure 4) “The kinase domains of ATM (B) and ATR (C)...” should be corrected to “The kinase domains of ATM (B) and ATR (D)”

The correction has been made.

- (Figure 5) Since the screen was performed to identify alleles sensitive in MMS or HU, and also since the western blot shows difference in Rad53 activation between MMS or HU among some alleles, it might be informative to display which mutants were identified to be sensitive in MMS and/or HU.

In our experience, the initial phenotypes can sometimes be misleading. This is likely due to the large scale-nature of the screen and the fact that cells often acquire incidental mutations that impact the phenotype during the relatively long (~ six weeks) screening process. For this reason, all our analyses were performed in a set of freshly derived strains in an otherwise WT background. However, the reviewer raises an important issue; we address this in a new section entitled

“Differential impact of *mec1* mutations on HU- versus MMS- dependent Rad53 activation and resistance to the stress” (lines 212-235).

- (Figure 5h) It is difficult to tell which mutant label refers to which spots on the sensitivity assay. There appears to be two replicates for each, a black line to connect the two could be placed, with the mutant labels above it. Additionally, the MMS images appear pixelated and blurry. A higher resolution image should be provided.

We provide a set of new images for Fig 5D.

- (Figure 5h) What is the genetic background of the *MEC1* deletion strain used as a control in the sensitivity assay? Since it is essential, there must be an additional suppressor mutation.

Both *mec1Δ* and *mec1-kd* are in a *smf1Δ* suppressor mutation background. This information is provided in Fig 5D and other relevant parts in the manuscript.

- (Figure 5h) Do any of these mutant alleles exhibit a growth defect in just unperturbed/no drug?

From the sensitivity assay, it appears not, but it is hard to tell. For example, C1985Y could exhibit some sickness, also considering that it is temperature sensitive, but it could also be due to uneven spotting of the culture.

The alleles that show modest but reproducible growth defect under unperturbed conditions (i.e. 23°C in the absence of any exogenous stress) are C467Y, A775P, G1124N, and G1804N. These alleles also confer the strongest temperature sensitive phenotype, which becomes apparent at 30°C (Fig 5D).

- *(Figure S2) Although this is out of the scope of the paper, how do the 24 / 50 mutations identified in the screen, that are not missense mutations, compare with the non-missense mutations identified in cancer-associated mutations of ATM/ATR? If they are similar or also conserved, this would further support Mec1 as a good model to understand ATM/ATR cancer mutations. In addition, it would be nice to see this available data, instead of only the missense mutation data. Together, all mutations identified would provide a nice data set for the literature.*

Of the 50 alleles sent out for sequencing, 24 came back as a WT MEC1 allele. We assume that this is the 24 the reviewer is referring to (apologies if this is not the case). It is likely that these 24 “mec1” strains carry a mutation elsewhere in the genome that confer HU/MMS sensitivity. Among the remaining 26 alleles, two carry the same truncation mutation at the 27th codon (Q27X), which would be a lethal allele. We suspect that these strains carry a second site suppressor mutation. To clarify, we provide this information in the Table 3 legend.

Reviewer #3 (Remarks to the Author):

The manuscript by Waskiewicz et al. presents analysis of mutations in ATM and ATR, master protein kinases involved in DNA replication and repair. They present results from three different aspects: 1) datamining of the cancer database 2) correlating these mutants with available structural information and 3) yeast genetic screens. The datamining has generated some useful results. The comparisons on the number and locations of the ATM and ATR mutations found in different cancer types are interesting. However the manuscript does not go beyond that and falls short in exploring the potential causes or the implications of these differences. For the second part the authors intended to relate these mutations to structures. This could have provided some useful insights. However, the authors simply mapped these mutations onto available ATR/Mec1 and ATM/Tel1 structures, all of which were at modest resolution, not taking advantage of all the recent structural information, including some high resolution structures. Further, a number of studies have already analysed in details of the potential structural consequences of some of the key cancer mutations, often as part of structural studies, the authors fail to quote these studies or to relate their analysis with those published. In addition to ATR and ATM, a number of other highly related PIKKs proteins such as Tel1, mTOR and DNA-PKc, have been reported and many key functional regions have been discussed in detail. The current study lacks sufficient depths in analysing the potential structural consequences of these mutations apart from whether they are in the active site, inside or on the surface of the structure.

A key objective of the manuscript was to gain a structural/function insight(s) into the numerous (~1,700) ATM/ATR residues mutated in cancer. Given the sample size, it was necessary to perform lower resolution analyses (e.g. Fig 3A-D). Our results provide insights that are different from and complementary to those obtained from some of the in depth analyses the reviewer mentioned. **However, we took the reviewer's concern on board and performed higher resolution analyses on a few selected residues (e.g. Fig 3E; Fig 4DE; Fig 6C, D, F). We also revised the Discussion to highlight implications of our findings.**

The genetic screens of mec1 identified some interesting mutations that render cells sensitive to HU/MMS agents which induces replication stress and DNA damage respectively. They measured levels of phosphorylated Rad53, as a measure of kinase activity, as well as cell viability. However, the results are presented sketchily, without detailed analysis of the relative activities and correlations. Furthermore, the results under HU and MMS challenges are not discussed separately nor the relationship between kinase activity and cell viability in details.

We added a section entitled "Differential impact of mec1 mutations on HU- versus MMS-dependent Rad53 activation and resistance to the stress" (lines 212-235) to address the reviewer's concern.

Overall the study is limited in scope and the manuscript presents limited new knowledge on ATR/ATM, some of the conclusions and statements in the manuscript are inaccurate or unjustified/supported. It would be important if the authors could conduct in-depth structure-function analysis with all available structural information of related PIKKs. Further, there are a significant amount of cancer mutations in ATM, and it would provide interesting and insightful knowledge if the authors carry out genetic analysis of Tel1 and/or conduct functional analysis of some of the interesting cancer mutations. Together, the new studies and analysis will provide insights into potential structural and functional consequences of these cancer mutations.

Unlike ATM, TEL1 is dispensable for normal proliferation and the DDR. Therefore, it is unlikely that genetic analysis of TEL1 will yield useful information unless the cell is in a mec1 background. Mec1 performs most functions of ATR and ATM; accordingly, it is utilized as a model for both. We provide this information in lines 190-7. We feel that "in-depth structure-function analysis with all available structural information of related PIKKs" would be beyond the scope of this manuscript.

Specific comments and suggestions:

1. The authors chose mutations that were selected based on conservation between species – this seems appropriate. However, sequence alignments between these proteins become difficult towards the N-terminal regions, due to species-specific insertions and variable numbers of HEAT repeats. Whilst the authors describe in the methods that they performed alignments in Clustal Omega, do other alignment

programs produce similar or different results? Does a structure-based sequence alignment also produce similar results? This is especially important as there are structures of both mammalian and yeast ATM and ATR, which would permit structure-based alignments to be made and therefore strengthen the comparisons made between yeast models and human disease mutations.

To address the reviewer's concern, we performed a comparative analysis of different sequence- and structure- based alignment programs. The finding is described in Supplementary Materials and Fig S1.

2. The unbiased genetic screen of *Mec1* is interesting and produces a number of mutations in the *MEC1* gene that reduce viability when challenged by genotoxic agents or heat shock. Importantly, the authors should demonstrate that each strain show similar expression levels of *Mec1* at the protein level.

Mec1 is an essential protein. Therefore, the fact that all our mutants are viable is an unequivocal genetic evidence that the protein is present at a sufficient quantity. However, we agree with the reviewer that the protein levels must be directly assessed.

Currently, there is no antibodies against *Mec1*. We tested published *HA-MEC1* and *MYC-MEC1* strains and found that they exhibit modest HU sensitivity, indicating that the tags impact protein function and/or stability. Given this, we are reluctant to utilize a tag because the results are likely to be misleading. *Mec1* antibody-production has been one of our top priorities for some time. We have raised and tested four polyclonal antibodies against two *Mec1* peptides but none was found to be specific to *Mec1*. We plan to try again taking advantage of the available structural information. We will perform the experiment as soon as suitable antibodies become available.

Clearly the *Mec1*-delta and the kinase-dead strains are viable and show similar sensitivity to other mutants when challenged with MMS, HU, but not heat. This is an important control to confirm that the mutations isolated do not result in loss of protein due to destabilisation of the structure leading to proteasome-mediated degradation, for example.

Both *mec1 Δ* and *mec1kd* are lethal mutations and are in a *sm1 Δ* suppressor mutation background. In contrast, all our mutants are viable in the absence of any suppressor mutation, indicating that all our mutant proteins are sufficiently stable and retains the intrinsic kinase activity required for viability. We included the status of *SML1* in Fig 5D to clarify this point.

Additional consideration should be given to whether mutations alter nuclear localisation of the protein kinase, which would also produce phenotypes similar to those presented in the screen if not allowed to localise correctly.

We agree with the reviewer. We plan to assess the impact of mutations on *Mec1* localization as soon as a suitable *Mec1* antibody becomes available (see above).

3. Some of the conclusions/statements are inaccurate or unjustified. Below are a few (but nonexclusive) examples: "only a small fraction of the critical residues is in the active site of the respective enzyme complexes, suggesting that loss of the intrinsic kinase activity is rare in carcinogenesis" "This sentence/conclusion is not justified/supported by the results presented in the manuscript.

We replaced the original low resolution images (Fig 4B, D) with higher resolution images (Fig 4D,E; Fig S4). These images show that majority of the residues are sufficient distance away (>20Å) from the catalytic center to directly impact the catalysis (e.g. Fig S6A).

"under unchallenged conditions, the dimeric complex exists as a minimally active enzyme, in which substrate-access to the active site is sterically hindered by several conserved features, including the PIKK-regulatory domain (PRD) in the kinase domain" This is not strictly true for *ATR/Mec1*.

To the best of our knowledge, what we wrote is the generally accepted view in the field. It is unclear to us which aspect of the sentence the reviewer is referring to as "not strictly true".

Page 5 line 130: "is buried inside the enzyme complex, implicating a structural or regulatory role(s). "This is not necessarily true as the only structures available are in autoinhibited state. These residues might be exposed or important in other functional states.

The reviewer is correct that the ATM and Mec1 structures we used are in an auto-inhibited state. However, the ATR is an active enzyme and the structure shows that majority of the ATR residues implicated in cancer are also buried inside (Fig 3C; Fig S3B). However, the reviewer raises a valid point and we explicitly state the possibility that some of the buried residues might become solvent accessible following enzyme activation (lines 339-40).

Furthermore, the majority of residues within a protein are inside, thus the percentage of mutant residues are probably proportional to the percentage of surface residues within a structure and this observation is thus expected.

We are somewhat confused with the statement that **“the majority of residues within a protein are inside” as this will depend on the protein. Moreover, we are simply stating the observation without** implying that this was either expected or unexpected.

Similarly, in terms of activation loop and catalytic loop, the size (< 10% of the kinase domain) is roughly in proportion to the relative number of mutations within these active site (~ 6 %). What is interesting is that mutations are disproportionally concentrated in the kinase domain (42% and 26%) which only occupies 10-15% sequence, suggesting many of these mutations are likely to modulate kinase activities, probably via altered regulation instead of catalytic activity per se.

We agree with everything the reviewer wrote here. We believe that the manuscript is written in a way to reflect this view. To further convey the point, **we incorporated the reviewer's words “likely to modulate kinase activities, probably via altered regulation instead of catalytic activity per se” in lines 172-173.**

REVIEWERS' COMMENTS:

Reviewer #1 (Remarks to the Author):

The authors have identified conserved, cancer associated mutations in DDR protein kinases ATM and ATR (from the cbioportal database) and mapped them onto available structures. They carry out an unbiased mutagenesis screen and show that several of the mutated residues correspond to conserved, cancer mutated residues in ATM and ATR and carry out preliminary functional assays on these mutants. I think that the work will be interest to those studying the effect of mutation in ATM/ATR in cancer.

Major comment:

I am unable to accept the authors argument stated on lines 148-150: that "The fact that majority of the solvent accessible ATM/ATR residues examined here are separated by a distance substantially greater than 13Å (Fig 3F) suggests that each residue may interact with a different protein(s)" .

While their statement on line 143 "that it is likely that many of the residues on the surface may mediate protein-protein interactions" may be correct, I do not believe that the authors can make the argument stated above.

I suggest modifying the first statement to " it is POSSIBLE that SOME of the residues on the surface may mediate protein-protein interactions" and request that the authors remove the second statement all together.

Minor comments:

There are several spelling mistakes and the word the is either missing or inappropriately used in several places in the manuscript.

Reviewer #2 (Remarks to the Author):

My comments have all been addressed. I particularly appreciate the new data on Rad53 activation vs damage sensitivity that is now included. I don't have any further comments.

Reviewer #3 (Remarks to the Author):

The manuscript has improved substantially with improved figures, analysis and text revisions. However, there are still a few issues that are not addressed adequately or not at all and some of the conclusions/sentences are not fully justified.

For example, their 'molecular' modelling was based on the low-resolution human ATM structure, which is limited and may be inaccurate in places. Indeed, several higher resolution structures of ATM and Tel1 are available, including a higher resolution map and model of human ATM (Xiao et al., 2019; Cell Research - PDB 6K9L) and an even higher resolution structure of Tel1 (Yates et al. 2020, Structure - PDB 6S8F). Since the authors here have focused on conserved residues between Tel1 and ATM, the Tel1 structure could be used for more accurate modelling and mapping the mutations onto the structure and to provide insights into potential structural and mechanistic consequences.

In fact, in the Yates et al. paper, many of the ATM cancer mutations were tabulated and mapped onto their structure, with some overlapping with some of the analysis carried out in this study. The authors should comment on the other study and compare the similarities and differences with the results in that study.

The resolution of the cryo-EM structures which the molecular models are built upon should be made obvious in figures, figure legends and methods, so that readers are aware of the potential limitations/caveats in the models and subsequent conclusions.

The figure could be made clearer, for example:

Figure 4, the colour of the PRD and FATC are too similar and hard to distinguish and should be changed. Furthermore, in various figures (Figure 4 and 6), additional functional domains/elements within the kinase domain, e.g. glycine-rich loop, α 3-helix, should also be highlighted in order to orientate readers to where the mutations are located in relation to key functional domains.

Figure 4 (f + g) and Figure 6(d-h), it would be helpful to orientate readers by adding a representation of the full protein structure with the regions being zoomed into highlighted or boxed.

Reviewer #1:

Remarks to the Author:

The authors have identified conserved, cancer associated mutations in DDR protein kinases ATM and ATR (from the cbioportal database) and mapped them onto available structures. They carry out an unbiased mutagenesis screen and show that several of the mutated residues correspond to conserved, cancer mutated residues in ATM and ATR and carry out preliminary functional assays on these mutants. I think that the work will be interest to those studying the effect of mutation in ATM/ATR in cancer.

Major comment:

I am unable to accept the authors argument stated on lines 148-150: that "The fact that majority of the solvent accessible ATM/ATR residues examined here are separated by a distance substantially greater than 13Å (Fig 3F) suggests that each residue may interact with a different protein(s)" .

While their statement on line 143 "that it is likely that many of the residues on the surface may mediate protein-protein interactions" may be correct, I do not believe that the authors can make the argument stated above.

I suggest modifying the first statement to " it is POSSIBLE that SOME of the residues on the surface may mediate protein-protein interactions" and request that the authors remove the second statement all together.

We amended the text as suggested and removed the second statement.

Minor comments:

There are several spelling mistakes and the word the is either missing or inappropriately used in several places in the manuscript.

We went through the manuscript and corrected all spelling and grammatical errors that we could find.

Reviewer #2:

Remarks to the Author:

My comments have all been addressed. I particularly appreciate the new data on Rad53 activation vs damage sensitivity that is now included. I don't have any further comments.

Reviewer #3:

Remarks to the Author:

The manuscript has improved substantially with improved figures, analysis and text revisions. However, there are still a few issues that are not addressed adequately or not at all and some of the conclusions/sentences are not fully justified.

For example, their 'molecular' modelling was based on the low-resolution human ATM structure, which is limited and may be inaccurate in places. Indeed, several higher resolution structures of ATM and Tel1 are available, including a higher resolution map and model of human ATM (Xiao et al., 2019; Cell Research - PDB 6K9L) and an even higher resolution structure of Tel1 (Yates et al. 2020, Structure - PDB 6S8F). Since the authors here have focused on conserved residues between Tel1 and ATM, the Tel1 structure could be used for more accurate modelling and mapping the mutations onto the structure and to provide insights into potential structural and mechanistic consequences.

We took the reviewer's concern on board, and repeated the analysis presented in Figure 3 (5.70 Å resolution ATM model, PDB 5NPO) in the higher resolution models of ATM (PDB 6K9L, 4.27Å) and Tel1 (PDB 6S8F, 4.00Å) (Supplementary Figure 3). The analysis shows the ATM residues mutated in cancer localize to essentially the same locations in the two ATM model structures (Supplementary Figure 3a-c). While there are some differences they are within the limit of resolution. (Supplementary Figure 3d). Accordingly, we do not believe that utilization of the 5.70Å resolution model, instead of the 4.27Å, would impact the key conclusions that majority of the ATM residues mutated in cancer are internally buried and distal to the active site.

In fact, in the Yates et al. paper, many of the ATM cancer mutations were tabulated and mapped onto their structure, with some overlapping with some of the analysis carried out in this study. The authors should comments the other study and compare the similarities and differences with the results in that study.

Our analysis shows that there are several hundred ATM residue mutated in cancer that are conserved in Tel1. In the Yates et al, the authors mapped just 17 of these residues onto a Tel1 model. To the best of our knowledge, the authors do not provide any information on how these 17 residues were selected from the several hundreds. It is possible that the residues were chosen because they happen to support their model. As such, we feel that we are unable to make a meaningful comment.

The resolution of the cryo-EM structures which the molecular models are built upon should be made obvious in figures, figure legends and methods, so that readers are aware of the potential limitations/caveats in the models and subsequent conclusions.

This is a good suggestion. We have provided the PDB and resolution information for all model structures in the figures and/or legends.

The figure could be made clearer, for example:

Figure 4, the colour of the PRD and FATC are too similar and hard to distinguish and should be changed. Furthermore, in various figures (Figure 4 and 6), additional functional domains/elements within the kinase domain, e.g. glycine-rich loop, α 3-helix, should also be highlighted in order to orientate readers to where the mutations are located in relation to key functional domains.

We tried incorporating additional structural elements to the Figures reviewer mentioned. Unfortunately, we found it to make the figures too complicated to comprehend as they are already highly content rich. We also tried manipulating different colour-combinations but didn't find them to be effective. We note that neither of the other two referees commented on quality of these figures.

Figure 4 (f + g) and Figure 6(d-h), it would be helpful to orientate readers by adding a representation of the full protein structure with the regions being zoomed into highlighted or boxed.

This is a good suggestion. We have provided a suitable full protein structure in Figures 4f, 6d, and Supplementary Figure 4c.